# Moonlighting Crypto-Enzymes and Domains as Ancient and Versatile Signaling Devices

**DOI:** 10.3390/ijms25179535

**Published:** 2024-09-02

**Authors:** Ilona Turek, Aloysius Wong, Guido Domingo, Candida Vannini, Marcella Bracale, Helen Irving, Chris Gehring

**Affiliations:** 1Australian Centre for Disease Preparedness, Commonwealth Scientific and Industrial Research Organisation, East Geelong, VIC 3220, Australia; ilona.turek@csiro.au; 2Department of Biology, College of Science, Mathematics and Technology, Wenzhou-Kean University, Wenzhou 325060, China; alwong@kean.edu; 3Wenzhou Municipal Key Lab for Applied Biomedical and Biopharmaceutical Informatics, Wenzhou 325060, China; 4Zhejiang Bioinformatics International Science and Technology Cooperation Center, Wenzhou-Kean University, Wenzhou 325060, China; 5Biotechnology and Life Science Department, University of Insubria, 21100 Varese, Italy; g.domingo@uninsubria.it (G.D.); candida.vannini@uninsubria.it (C.V.); marcella.bracale@uninsubria.it (M.B.); 6La Trobe Institute of Molecular Sciences, La Trobe University, Bendigo, VIC 3552, Australia; 7Holsworth Initiative for Medical Research, Rural People, Department of Rural Clinical Sciences, La Trobe Rural Health School, La Trobe University, Bendigo, VIC 3552, Australia; 8Department of Chemistry, Biology and Biotechnology, University of Perugia, 06121 Perugia, Italy

**Keywords:** crypto-domains, crypto-enzymes, proteomes, adenylate cyclase, guanylate cyclase, phosphodiesterase, heme-proteins, H-NOX, plant hormones, abscisic acid (ABA)

## Abstract

Increasing numbers of reports have revealed novel catalytically active cryptic guanylate cyclases (GCs) and adenylate cyclases (ACs) operating within complex proteins in prokaryotes and eukaryotes. Here we review the structural and functional aspects of some of these cyclases and provide examples that illustrate their roles in the regulation of the intramolecular functions of complex proteins, such as the phytosulfokine receptor (PSKR), and reassess their contribution to signal generation and tuning. Another multidomain protein, *Arabidopsis thaliana* K^+^ uptake permease (AtKUP5), also harbors multiple catalytically active sites including an N-terminal AC and C-terminal phosphodiesterase (PDE) with an abscisic acid-binding site. We argue that this architecture may enable the fine-tuning and/or sensing of K^+^ flux and integrate hormone responses to cAMP homeostasis. We also discuss how searches with motifs based on conserved amino acids in catalytic centers led to the discovery of GCs and ACs and propose how this approach can be applied to discover hitherto masked active sites in bacterial, fungal, and animal proteomes. Finally, we show that motif searches are a promising approach to discover ancient biological functions such as hormone or gas binding.

## 1. What Are Crypto-Enzymes

Crypto-enzymes or cryptic enzymes are proteins with one or several hidden enzymatic functions [1]. Some of these functions have remained hidden, even in the genomics era, and the reasons are two-fold. Firstly, amino acid sequences that can perform distinct catalytic processes have not been apparent in sequence comparisons because they may have been the result of convergent evolution and are insufficiently similar in their amino acid composition to known enzymes and have hence not been detected [2]. An alternative explanation could be that the minimal number of amino acids required for catalysis is too small to be identified with a reasonable degree of confidence with sequence comparison tools such as BLAST. Such cryptic enzymes (also referred to as crypto-enzymes) have rekindled our interest in the question of the amino acid requirements in catalytic centers and the requirements of protein folding essential for enzymatic function [3]. Both the presence of key amino acids in catalytic centers that are essential and sufficient for catalysis and the structural conditions necessary for catalysis have had key roles in the discovery of cryptic enzymes in plants [4,5,6,7].

A further general aspect of cryptic enzymes in plants is functional. These enzymes may have much lower activities as compared to their canonical equivalents. This, however, may precisely be their advantage since a lower activity can in some cases be not just desirable but in fact an essential condition for their function in vivo [4,8,9]. A case in point involves the cryptic guanylate or adenylate cyclases where catalytic products cGMP or cAMP, respectively, regulate other intramolecular functions such as those of kinases [10,11,12]. In such a case, the catalytic product may act on neighboring catalytic centers in other domains of the same protein. Such dual or multiple activity proteins are sometimes termed “moonlighting” proteins [13,14]. Hidden catalytic sites in moonlighting proteins and their products may also contribute to crowded microenvironments within their vicinity [15,16,17]. Given that, e.g., cGMP and cAMP affect many processes including protein phosphorylation [18,19], we would argue that the cryptic catalytic activities of GCs and ACs safeguard against signal flooding that might well cause undesirable unspecific responses.

## 2. Uncovering Crypto-Enzymes in Moonlighting Proteins—Plant Guanylate Cyclases as a Case in Point 

In plants, responses to 3’,5’-cyclic monophosphates were initially a subject of controversy [20] but have now been established [21,22,23,24,25,26,27,28,29,30,31,32]. Notably, the discovery that the regulation of ion transport is in part cGMP-dependent [33,34] brings to the forefront the roles of specific cyclic nucleotides. This was first established by pioneering electrophysiological studies in the 1990s that demonstrated that in *Arabidopsis thaliana*, voltage-dependent hyperpolarization-activated K^+^ current gating could be achieved by 3′,5′-cGMP but not 3′,5′-cAMP [33]. Others reported that cAMP could indeed stimulate *Vicia faba’s* mesophyll K^+^ channels [35]. Consistent with these findings, it was subsequently reported that kinetin and plant natriuretic peptide (PNP)-dependent stomatal opening [36] requiring K^+^ uptake is dependent on cGMP [37,38]. These and an increasing number of other reports linking, e.g., cGMP to light perception and phytochrome-dependent signaling [39,40,41] spurred the search for plant cyclases and guanylate cyclases (GCs) in particular. It was perhaps surprising that the release of the *A. thaliana* draft genome in 2000 [42] did not identify any obvious candidate GCs since queries with annotated orthologs from bacteria, fungi, and vertebrates did not point to any plausible orthologs [2]. This could be explained by either an unusually high level of divergence in plant GCs that has put them outside the detection limit of BLAST searches or by the fact that plant GCs may not be homologous to bacterial, algal, fungal or animal GCs. In both cases, however, we hypothesize that plant GCs may contain a high degree of similarity in the amino acid composition in the catalytic center with established GCs. With this in mind, functionally assigned amino acids from the designated catalytic centers [43,44,45,46,47,48] from vertebrates and prokaryotes were aligned with a view to deduce a GC search term. Such a search term or motif was then used to query the complete *A. thaliana* genomic sequence to identify candidate proteins that contained the motif. The candidates were then evaluated structurally by modelling and substrate-docking studies [49], and promising candidates were chosen for extensive functional testing in vitro and in vivo. 

With the discovery and testing of the first functional GC found in higher plants based on catalytic center motifs, it was noted that this was potentially a powerful approach to find other GCs, in plants and indeed other organisms, particularly if the entire proteome of the species of interest was in the public domain. It was also noted that GCs, including the plant GCs, occur in complex proteins with varied domain organizations [5], and this in turn points to complex functions. In parallel with the characterization of candidate GCs found with the original motif, the search motif itself was modified with a view to discover the entire complement of GC catalytic sites in whole proteomes. The changes to the motif, i.e., expansion and/or increasing or decreasing the stringency, were based on theoretical considerations, such as the addition of structurally similar amino acids to the motif, and were supported by site-directed mutagenesis and in vitro GC activity testing of mutations in catalytic centers [50,51,52]. Structural modelling including substrate docking has proven to be an invaluable additional tool for the assessment of candidate GCs [6,7,53,54]. It appears that there is still a substantial number (>40) of multidomain molecules with potentially functional cryptic GC catalytic centers in the *A. thaliana* proteome that remain to be characterized. We have also noted that in orthologs of already tested GCs in *A. thaliana*, the degree of conservation is particularly high in the catalytic centers. While not surprising, this adds confidence to the assessment of the functional importance of these moonlighting crypto-enzymes.

## 3. Cryptic Moonlighting Enzymes in Animal Proteomes—Uncovering GCs in Humans

Discoveries in the last two decades have established an ever-increasing number of functional GCs moonlighting catalytic centers in plant proteomes. Many of them nested within kinase domains where their catalytically generated cGMP has proven essential for both intramolecular and downstream signaling. Given the success of discovering functional cryptic sites with targeted motif searches, this method has also been applied to other proteomes and notably the proteome of *H. sapiens*. In the human proteome, 18 candidates were identified, including interleukin-1 receptor-associated kinase 3 (IRAK3; Q9Y616) [55] and neurotropic receptor tyrosine kinase 1 (NTRK1; P04629) [56]. The identification of several proteins previously established as GCs (e.g., atrial natriuretic peptide receptor 1 and 2) [57] among these 18 candidates increased confidence in the validity of the search. Of importance, IRAK3 and NTRK1 were the only members of the IRAK or NTRK family that were identified as putative GCs. In fact, the immune checkpoint protein IRAK3 was the first characterized animal crypto-GC capable of generating similar amounts of the cGMP product to those produced by the plant phytosulfokine receptor [55]. The functional GC center of IRAK3 is embedded in its pseudokinase domain, and the preferred cofactor for cGMP synthesis is Mn^2+^. Importantly, mutations in and nearby the GC center not only modify the capability of IRAK3 to produce cGMP but also have implications for IRAK3-dependent downstream innate immune signaling, including suppression of the lipopolysaccharide-induced activation of NFκB resulting in the downregulation of inflammation by IRAK3 [58,59]. Certain mutations within the GC center were reported to affect downstream cytokine production and change the subcellular localization of IRAK3, likely due to an altered IRAK3 interactome [59]. Furthermore, while complementation of IRAK3-knockdown cells with wild type IRAK3 inhibits cytokine generation, complementation with IRAK3 mutants within the GC center only partially restores cytokine production [8,59]. NTRK1 shows a domain architecture much like that of plant receptor kinases such as the phytosulfokine receptor, where a functional GC essential for downstream signaling is embedded within a kinase domain. Similarly to IRAK3, NTRK1 shows catalytic activity in vitro and a marked preference for Mn^2+^ over Mg^2+^. This, therefore, points to hitherto unsuspected roles of cGMP in the intramolecular and downstream signaling of NTRK1 and the role of cGMP in NTRK1-dependent growth and neoplasia. Finally, it appears that the catalytic center-based searches for hidden mononucleotide cyclases in animal proteomes promise a raft of new discoveries and insights into biological signaling as previously suggested [60]. 

## 4. Structural and Functional Aspects of Cryptic Mononucleotide Cyclases

There are many different combinations of functional domains that combine with cyclic nucleotide cyclase centers within complex proteins [5,61,62,63]. The different domains include gas-binding domains, protease domains, phosphodiesterase (PDE) domains, and kinase domains, and these domains are influenced by the catalytic products of the cyclases in different ways and to various degrees, also depending on other cellular components such as pH and Ca^2+^ concentration. Here we shall again concentrate on plant proteins where, for instance, AC catalytic centers affect the response to pathogens and gravitropism [12,64]. The proteins that harbor both kinase and GC domains stand out as particularly numerous and constitute a distinctive subset of the leucine-rich repeat receptor-like kinase protein family [65,66]. The latter includes a brassinosteroid receptor (BRI1) [50,67,68,69], a phytosulfokine receptor (PSKR1) [51,70,71,72], plant elicitor peptide receptor 1 (PEPR1) [73], and plant natriuretic peptide receptor 1 (PNPR1) [74]. Given the number of proteins discovered that contain these domain combinations, questions arise such as how these structural features are reflected in the function of the protein, whether crosstalk exists between the different activities, and whether there are other messengers such as cytosolic Ca^2+^ signatures [71] that modulate these activities. Additional consideration should be given to the cellular environment in which these crypto-enzymes within complex proteins operate. The cytoplasm is a crowded place where scaffold proteins have evolved to keep key enzymes in the correct place to ensure ordered spatial, temporal, and stimulus-specific message generation and processing [75,76], and it is conceivable that complex moonlighting proteins and crypto-enzymes are uniquely suited to perform in dense cellular microenvironments [15,16]. 

Similar questions arise in the case of the dual (GC/AC)/PDE domain-containing proteins [77,78] where the cyclic mononucleotide-generating and hydrolyzing enzymes reside within the same moonlighting protein. Such a protein architecture would enable these molecules to operate as “dimmer switches” where the AC or GC generates the “on” signal within the microenvironment and the PDE operates the “off” or damping signal [79]. The elucidation of the mechanisms that govern the different enzymatic activities, as well as the nature of other components such as Ca^2+^, calmodulin, and fluctuating pH [80,81,82] within the cellular (micro-)environment that all have a role in these processes, promises new insights into signal generation and tuning.

## 5. Phytosulfokine Receptors Harbor Hidden Nucleotide Cyclases 

Phytosulfokines (PSKs) are sulfated plant pentapeptides that stimulate growth and modulate differentiation and defense responses [83]. PSKs act through a dedicated receptor, PSKR1 [84,85]. PSKR1 was one of the early plant leucine-rich repeat receptor-like kinases discovered that contains a cytoplasmic GC domain [15,51]. Curiously, the GC catalytic center in PSKR1 is entirely embedded within the kinase domain (Figure 1), which might suggest that there may be a tight functional link between the two catalytic activities that would in turn affect downstream PSK signaling. Transient overexpression of PSKR1 in leaf protoplasts of *A. thaliana* caused a 20-fold cGMP rise, demonstrating that the receptor has GC activity. The response has also been shown to be specific since it is elicited by biologically active sulfated PSK-α but not by the un-sulfated peptide backbone [51]. These findings show that the receptor not only contains a functional kinase but also within it a nested active GC, making PSKR1 a member of an entirely novel class of enzymes characterized by overlapping active catalytic domains. The first question that arises is whether kinase and GC activities are independent of each other or if there is a switch that regulates the activities. In answer to this question, it was shown that mutations within the GC catalytic center reduce GC activity without affecting kinase catalytic activity [71] and impair seedling growth [86]. Interestingly, Ca^2+^ levels within the physiologically relevant range enhance GC activity more than twofold in a concentration-dependent manner. Conversely, increased Ca^2+^ levels strongly inhibit kinase activity up to 500-fold at 100 nM Ca^2+^ [71]. These effects imply that Ca^2+^ at physiological levels is the bi-modal switch between the kinase and GC activities of PSKR1. Interestingly, the PSKR1 receptor forms a multi-protein complex at the plasma membrane with BAK1 [87], H^+^-ATPase, and cyclic nucleotide-gated channel 17 (CGNC17) [86], raising the possibility that cGMP generated by PSKR1 regulates CNGC17 activity or indeed ATP-ases [88].

A further level of regulation was proposed based on the observation that cGMP itself strongly inhibits kinase activity [11]. It is conceivable that cGMP and PSKR1, BRI1 or PEPR1 form a complex activating negative feedback mechanism(s) and leading to the desensitization of the receptor and hence the suppression of cGMP-dependent pathways. In this model (Figure 2A), kinase activity is the default signal and ligand activation triggers cGMP production and downstream effects. In such a case, comparatively low but highly localized cGMP generated by the GC in response to a ligand is sufficient to modulate the kinase activity [16]. This type of receptor therefore provides a switch within a single molecule that can directly and/or indirectly affect ligand-dependent phosphorylation in downstream signaling cascades. To gain further insight into the tuning of the overlapping dual catalytic function in vitro and modes of crosstalk, additional analytical tools were employed. First, tandem mass spectrometry revealed at least 11 phosphorylation sites (eight serines, two threonines, and one tyrosine) within the cytosolic domain of the PSKR1 [70,89]. Subsequent phosphomimetic mutations of three serine residues (Ser686, Ser696, and Ser698) in tandem at the juxta-membrane position caused enhanced kinase activity in the “on” mutant and suppression in the “off” mutant. However, both types of mutations reduced guanylate cyclase activity (Figure 2B). In turn, “on” and “off” phosphomimetic mutations of the phosphotyrosine (Tyr888) residue in the activation loop suppressed kinase activity, while neither mutation affected guanylate cyclase activity (Figure 2B). 

Size exclusion and analytical ultracentrifugation analyses of the recombinant cytoplasmic domain of PSKR1 suggest that the receptor protein is reversibly dimeric in solution, and this was further confirmed by bimolecular fluorescence complementation [70]. Taken together, these experiments suggest that in this novel type of receptor architecture, specific phosphorylation and dimerization are likely essential mechanisms for ligand-mediated catalysis and signaling.

## 6. A K^+^ Channel with Multiple Crypto-Domains 

Further cryptic moonlighting enzymatic activities are found in the *A. thaliana* K^+^ uptake permease (AtKUP5) (Figure 3A). 

AtKUP5 is a complex cation channel protein with at least two spatially and catalytically distinct enzyme activities, a cytosolic N-terminal AC and a cytosolic C-terminal PDE [61,78,90]. The quest for the discovery of plant ACs in proteomes received serious impetus when the well-tested GC motif was modified in the amino acid responsible for substrate recognition [64,91]. AtKUP5, one of the recently discovered *A. thaliana* ACs, is a K^+^ uptake permease with a catalytic site embedded in its N-terminal cytosolic domain [90]. 

K^+^ flux requires the presence of the canonical amino acids of the functional AC. Site-directed mutagenesis of amino acids in the AC catalytic center that severely impair AC function is accompanied by significantly restricted K^+^ net fluxes. Perhaps surprisingly, AtKUP5-mediated K^+^ flux is not affected by cAMP, the catalytic product of the AC itself. However, K^+^ influx into the cytosol causes cAMP accumulation. It is therefore conceivable that AtKUP5 operates as a K^+^ flux sensor whereby increased K^+^ concentrations affect downstream responses via cAMP-dependent processes or components such as cyclic nucleotide-gated channels (CNGCs) [92,93,94,95]. 

It has long been understood that cyclic mononucleotide-dependent signaling would require not just generators of an “on” signal in the form of cyclases but also generators of the “off” signal, such as specific PDEs [78,79]. PDEs have remained somewhat elusive in plants, but again, consensus motifs based on comparative sequence analyses and structural features of annotated PDE catalytic centers allowed the building of amino acid search motifs for the identification of PDE candidates in proteomes in general and particularly in the proteome of *A. thaliana*. Perhaps surprisingly, one of the PDE candidates was AtKUP5, which also contains a functional AC essential for K^+^ transport/sensing (Figure 3A). 

Structural analyses and molecular docking revealed that the PDE domain occupies the C-terminal of protein, forming a distinct solvent-exposed pocket that can spatially accommodate the cyclic adenosine monophosphate (cAMP) substrate [78]. PDE activity was subsequently confirmed by liquid chromatography tandem mass spectrometry (LC–MS/MS). In addition, catalytic activity is stimulated by the Ca^2+^/calmodulin (CaM) complex. Taken together, AtKUP5 displays this dual cryptic enzyme AC-PDE architecture, which might enable the finetuning of cAMP concentrations within the orbit of the AtKUP5 channel and thereby regulating cAMP-dependent signaling that in turn maintains K^+^ homeostasis.

The F-box is a protein domain of about 50 amino acids with a role in protein–protein interactions [96,97]. Since such domains are essential for the formation of various biologically active complexes with signaling roles, it is perhaps not surprising that they occur in combination with other signaling components. In *A. thaliana*, F-boxes were predicted in proteins that also harbor AC domains [91], potentially linking specific F-box-dependent protein–protein interactions to cAMP-dependent signaling. These predictions were recently vindicated by a report of a moonlighting AC within the well-characterized transport inhibitor response 1/auxin-signaling F-box (TIR1/AFB) auxin receptors [12,30,61,98]. 

Closer inspection of another Arabidopsis F-box protein (At3g44080, see Figure 3B) is particularly interesting since it does not just contain an AC domain but also a GC domain as well as a phosphodiesterase (PDE) domain. If we assume that the AC, GC, and PDE are active, then a single protein could catalyze the reaction from ATP to cAMP and from GTP to cGMP, as well as hydrolyze these respective cyclic nucleotides to AMP and GMP. This would make such an F-box protein a tuner of cyclic mononucleotide downstream signaling. 

## 7. Cryptic Mononucleotide Cyclases Can Tune Signal Networks

First, the above examples of cryptic enzymes, notably GCs and ACs, suggest that they are essential parts of complex proteins where they act as tuners of other functions within these proteins and possibly other nearby protein complexes. Secondary messengers such as Ca^2+^ provide additional avenues in regulating the cryptic enzymes. Since many of these complex proteins have recognized signaling functions, e.g., as receptors, the catalytic activity of the cryptic enzymes modulates their signal output and hence the downstream signaling cascade. To identify targets within such cascades, several approaches have been adopted. They include phospho-proteomics to detect and quantify stimulus-specific cAMP- [18] or cGMP- [19] dependent phosphorylation events (Figure 4). 

It is noteworthy that cAMP-dependent phosphorylation targets are conserved among plant species, and this messenger profoundly affects the plant kinome; particularly mitogen-activated protein kinases (MAPKs), receptor-like kinases, and calcium-dependent protein kinases [18,29,99]. An interaction between the cAMP and MAPK signaling cascades was also observed by overexpressing the adenylate cyclase catalytic center of Arabidopsis AtKUP7 to increase cellular cAMP levels [18] (Figure 4).

This induction of protein kinases will, in turn, modulate the signaling cascade and thereby enable tuned cellular responses at the systems level. It was also observed that cAMP-dependent phosphorylation mainly occurs amongst RNA-binding proteins [100], including proteins with a role in RNA splicing, in particular serine/arginine-rich (SR) proteins, which are differentially phosphorylated in response to cAMP depletion. This is yet another indication that cAMP-dependent effects on mRNA processing may in turn have a profound effect on cellular programming in response to developmental and environmental cues [18]. 

Incidentally, cGMP also rapidly and specifically affects the kinome [19], and again, the altered phosphorylation signature points to a role in spliceosome assembly and function, and hence, systems-level changes.

Dynamic phosphorylation of spliceosomal proteins and accessory splicing factors is critical for the proper regulation of both constitutive and alternative splicing events [101,102]. The presence of both cyclic nucleotides was essential to regulate the phosphorylation status of elements involved in all crucial steps of spliceosome assembly and activation (Figure 4). In detail, cGMP-dependent changes in the phosphorylation status of U1, U4, U5, and U6 spliceosomal RNAs were reported, whereas cAMP depletion can lead to an altered regulation of the ribonucleoprotein U2 and some spliceosome-associated proteins (SAPs). UBP1-associated protein 2A (AT3G56860) was the only spliceosomal element commonly regulated by cAMP and cGMP.

A functional link between the cAMP-dependent signaling mechanisms and cytosolic Ca^2+^, Na^+^, and K^+^ fluxes has been established [103,104,105]. Interestingly, by searching the phospho-regulated proteins in cAMP-buffered tobacco BY-2 cells for AC catalytic centers with ([RKS][YFW][DE][VIL]X(8,9)[KR]X(1,3)[DE], [106]), this domain was detected in a K^+^ uptake transporter 7 (KUP7) paralog [18]. In *A. thaliana*, the N-terminal cytosolic region (1–100) of KUP7 contains an AC catalytic center able to generate cAMP [29], supporting the notion that this motif is indeed conserved in plants. Moreover, these findings point to a possible feedback regulation mechanism by which cAMP levels can be kept tuned.

The phosphorylation status is also essential for the activation of GC domain-containing kinases, such as PSKRs [51,70,89,107], whose transient over-expression increased cGMP levels in Arabidopsis leaf protoplasts [51]. BRI1 kinase activity is also influenced by its phosphorylation status. This regulation in turn enables a cascade of phosphorylation events that may regulate BRI1 substrate kinases (BSKs) [67,108,109].

Second, we can gain further insight into the role of crypto-enzymes by studying the interactome [110] of their catalytic products. Over 10 cyclic nucleotide-binding proteins including key enzymes in the Calvin cycle and photorespiration pathway were identified [110]. Eight of them contain annotated cyclic nucleotide-binding domains. The identified proteins are post-translationally modified by nitric oxide (NO), transcriptionally co-expressed, and annotated to function in hydrogen peroxide signaling and the defense response. It was therefore proposed that cAMP- and cGMP-binding proteins function as points of crosstalk between the cyclic mononucleotides, nitric oxide, and reactive oxygen species signaling during defense responses.

## 8. Ancient Mononucleotide Cyclases

Nucleotide signaling including cyclic nucleotide signaling in bacteria is well established (for a review see [111]). However, the identification and structural and functional characterization of ACs [104] and GCs [112] in prokaryotes appear to be much less straightforward. Furthermore, it is unclear how many different ACs or GCs are present in different bacterial proteomes. Inspection of the UniProt database would suggest that there is only one bona fide AC present in the *E. coli* (K12 strain) proteome, and this is cyaA (UniProt entry: P00936). In the case of the GCs, we found one (UniProt entry: A0A8T6BIH7) annotated as GC, one with AC/GC dual-activity (UniProt entry: A0A5J9X946), and one protein with an AC/GC domain (UniProt entry: A0A2I8SQF0) but none in strain K12. 

Given the somewhat surprising success to detect hidden functional ACs and GCs in the *A. thaliana* proteome, we decided to search the *E. coli* proteome for hidden candidate ACs and GCs using the search term [RKS][YFW][DE][VIL]X(8,9)[KR]X(1,3)[DE] for ACs and [RKS][YFW][DE][VIL]X(8,9)[KR]X(1,3)[DE] for GCs. The first term retrieved >25 proteins with hidden AC domains and the second term retrieved >70 hidden candidate GCs. A number of the retrieved candidate ACs and GCs are annotated as domains of unknown function (DUF) family proteins [113]. The AC search term also retrieved an EAL domain-containing protein (UniProt entry: Q8X6V4). EAL domain-containing proteins are conserved bacterial signaling proteins including PDEs with substrate specificity for cAMP [114]. Incidentally, the twin cyclic mononucleotide cyclase and PDE domain architecture in complex signaling proteins is not an uncommon feature in plants [77], possibly enabling such proteins to function as cyclic nucleotide signaling nodes since they can both generate and hydrolyze cyclic nucleotides. Parenthetically, the GC motif also retrieves a candidate GC annotated as 3′-cyclic-nucleotide 2′-phosphodiesterase/3′-nucleotidase (UniProt entry: Q8XCG7) [77]. Finally, an outer membrane transport protein (UniProt entry: Q8XD20) is also a putative GC, and this would constitute a similar domain combination as seen in two *A. thaliana* potassium transporters that are also nucleotide cyclases [90,106].

In the single-celled blue alga *Chlamydomonas reinhardtii*, there are 50 proteins with annotated AC domains and >60 proteins with annotated GC domains. It has previously been reported that GCs themselves come in >20 different domain combinations with >10 different partners, including H-NOX domains, GAF-like domains, CheY-like domains, and cysteine proteases [5,115]. This diversity was interpreted as indicative for the role of cyclases and their catalytic products in developmental responses and responses to the environment.

## 9. Finding Conserved Hidden Functions beyond Crypto-Enzymes

While consensus motif searches have been highly successful in the identification of cryptic enzymes such as GCs, ACs, and PDEs [61], particularly in whole-proteome searches, this approach can also be applied to peptidic moonlighting sites without catalytic functions [54,116,117,118,119,120]. Two recent examples are the H-NOX heme binding centers in gas-sensing and gas-responsive proteins [116,119,120,121,122]. It was long assumed that heme-based NO-sensing by H-NOX proteins was not a feature of higher plants. However, it turned out that NO-sensing H-NOX proteins are part of complex multi-domain proteins. In these proteins, only amino acids that directly participate in coordinating the heme moiety remain highly conserved, which made them undetectable by standard homology searches. Alignment of the heme core amino acid sequences of canonical H-NOX domains has indeed suggested the presence of highly conserved histidine (H) and proline (P), as well as the downstream tyrosine-X-serine-X-arginine (YXSXR) signatures-‘‘X’’ represents any amino acid-in a number of plant proteins. In plants, these amino acids are not only highly conserved but also coordinate the heme porphyrin moiety within the pocket, thus directly affecting NO binding [123,124]. In an attempt to identify plant heme-based NO sensors, an H-NOX core consensus sequence motif HX(12,14)PX(14,16)YXSXR, was deduced. It includes only the functionally annotated amino acids separated by gaps where numbers in parentheses represent the range of any amino acid X between conserved amino acids in the motif [121]. It appears that through divergent evolution, proteins have incorporated NO-sensing capabilities to yield domain combinations in addition to those of H-NOX-containing soluble GCs in vertebrates. Furthermore, an even more relaxed H-NOX motif may uncover further NO-sensing proteins, not just in plants, but across other kingdoms. 

The second example is the quest for abscisic acid (ABA)-binding and ABA-interacting proteins in plants and animals. The sesquiterpene ABA is an ancient stress response molecule that is best characterized in plants where many ABA-dependent processes operate via the PYR/PYL/RCAR receptor complexes [117,118,125,126]. Still, there is evidence that not all plant responses operate through this mechanism [117], and this is entirely consistent with ABA responses in animals and humans that do not appear to contain such canonical receptors. It was therefore hypothesized that plant and animal proteomes harbor cryptic undiscovered ABA-binding sites in complex moonlighting proteins that may have been acquired during divergent evolution. Such proteins could conceivably be identified with carefully curated amino acid search motifs deduced from the binding sites of canonical ABA receptors [117]. The first discovery enabled by this approach was that the guard cell outward-rectifying K^+^ channel (GORK) is responsive to ABA even in the absence of the canonical receptor, and this response was conditional on the ABA-binding motif. This implies that specific ABA binding can occur in hitherto unidentified ABA-binding proteins, several of which have since been modelled and assessed structurally to establish the presence of a fold compatible with ABA binding [117]. These bona fide ABA interactors imply that the regulation and tuning of ABA-dependent processes are considerably more complex than hitherto suspected. A case in point would be the experimental verification of a potential ABA binding site in the PDE domain of KUP5 (Figure 3A). Finally, further experimental proof of proposed ABA-binding targets in animals [127] will help to better understand the role of this ubiquitous hormone in processes such as immune responses and tumor progression in animals [128,129].

## Figures and Tables

**Figure 1 ijms-25-09535-f001:**
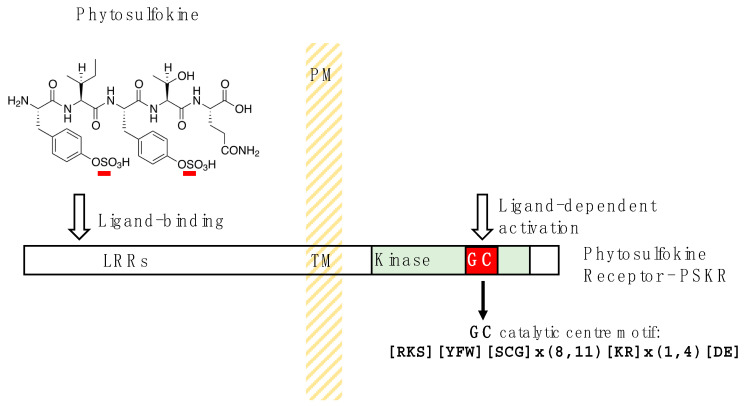
Domain organization of the phytosulfokine receptor (PSKR). In this leucine-rich transmembrane (TM) receptor kinase, the sulfonated ligand (S residues underlined in red) will, upon binding of the leucine-rich region (LRR), activate the guanylate cyclase (GC) nested within the cytosolic domain of the transmembrane (TM) receptor. PM stands for plasma membrane. The letters in square brackets [] represent different amino acids allowed in a position of the motif.

**Figure 2 ijms-25-09535-f002:**
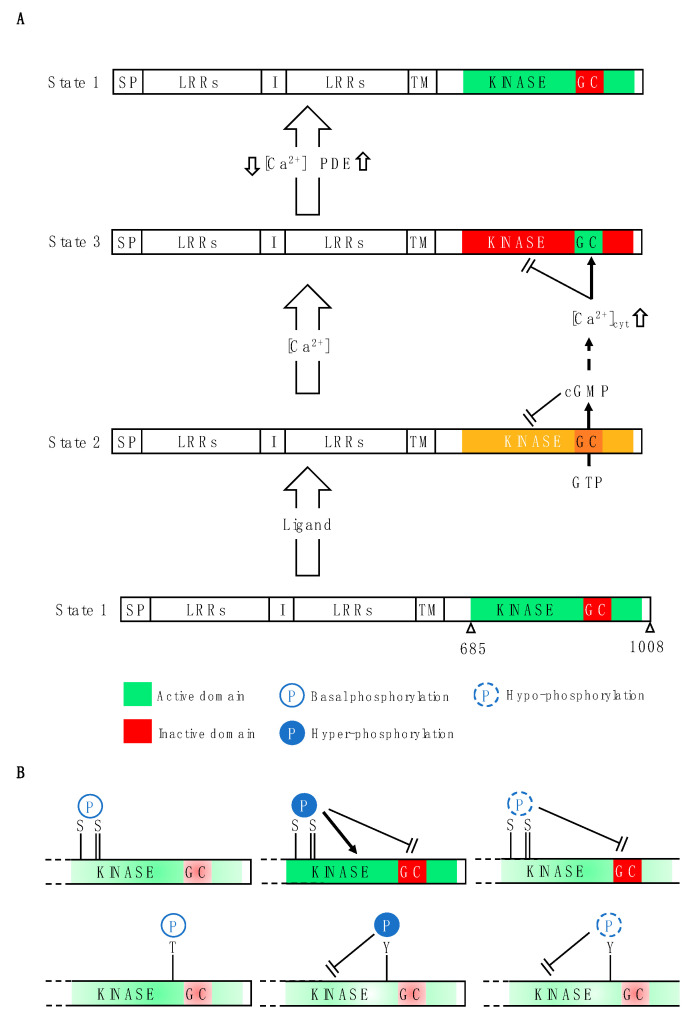
Mechanistic models of regulation of cryptic guanylate cyclase (GC) activity. (**A**) Several plant cryptic GCs are found embedded in kinase domains of leucine-rich repeat (LRR) receptors, where, in state 1, the GC is inactive while the kinase is active. In state 2, auto-generation of cGMP reduces kinase activity that is further inhibited by Ca^2+^ that subsequently activates the GC. Reducing cGMP by the action of phosphodiesterases (PDEs) and decreased cytoplasmic Ca^2+^ levels return the cryptic GC to state 1. (**B**) The phosphorylation state contributes to kinase and GC activities. Hyperphosphorylation at serine residues stimulates kinase activity while inhibiting GC activity, whereas altering tyrosine phosphorylation using permanently on or off mimetics inhibits kinase activity without affecting GC activity (for details see text and also [70]).

**Figure 3 ijms-25-09535-f003:**
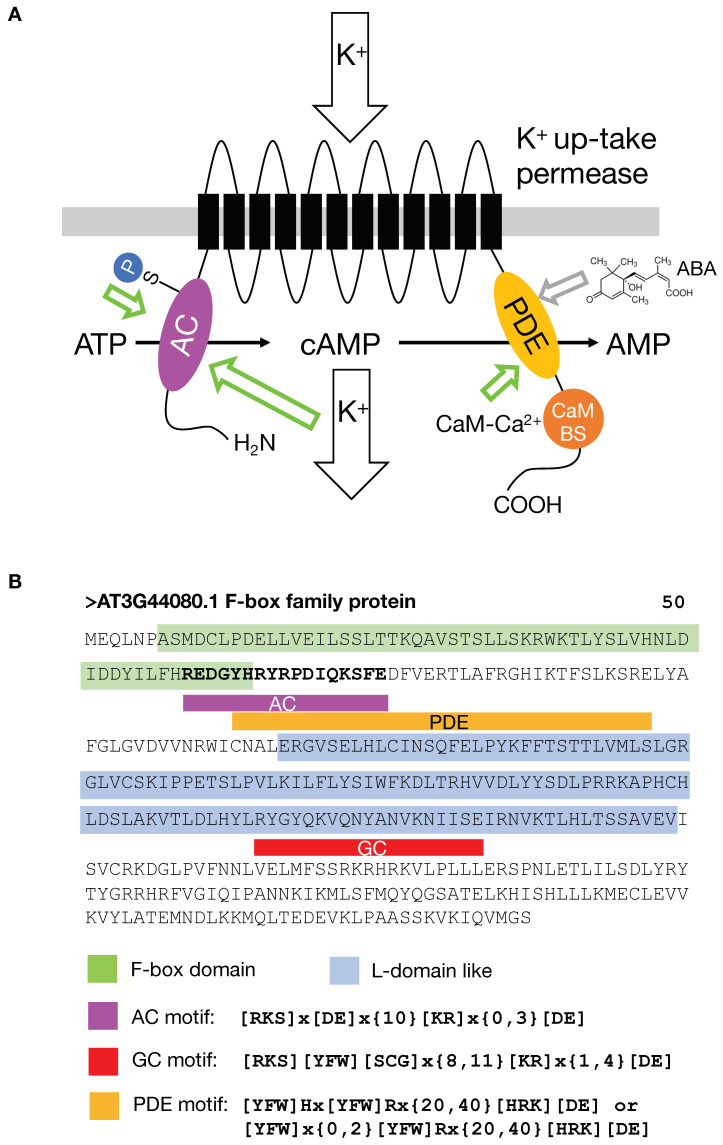
Dual crypto-enzymes. (**A**) Schematic of the architecture of AtKUP5 showing the intracellular N-terminal AC center that is modulated by K^+^ ions to generate cAMP and the PDE center that uses cAMP as its substrate and is activated by Ca^2+^-CaM binding to the CaM binding domain. The green arrow marks enhancing activities and the grey arrow points to a putative ABA binding site in the PDE domain, (**B**) Amino acid sequence of an *A. thaliana* F-box protein (At3g44080.1). The three domains that regulate the cyclic mononucleotide content in the cellular microenvironment are the AC, the GC, and the PDE. All three domains were identified with amino acid motifs based on functionally assigned common residues in the respective catalytic centers.

**Figure 4 ijms-25-09535-f004:**
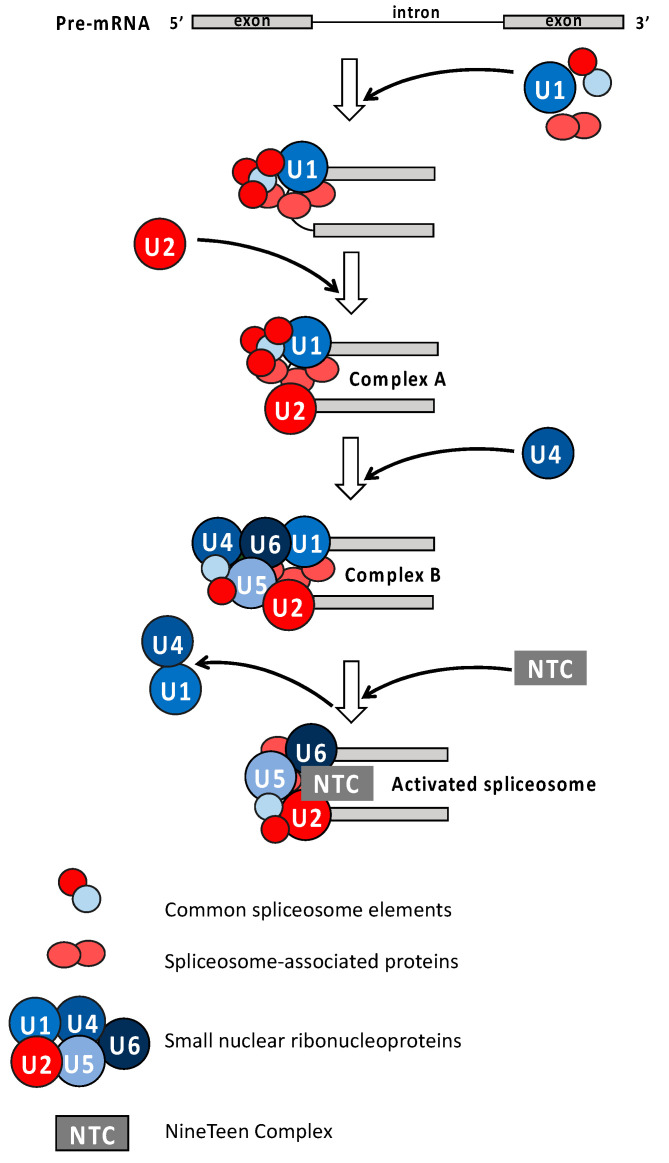
Schematic representation of spliceosome assembly and activation. Spliceosomal proteins and accessory splicing factors regulated by cAMP and cGMP are colored in different shades of red and blue, respectively.

## Data Availability

All data referred to in the manuscript are in the public domain.

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
