# Peer review of "Moonlighting Crypto-Enzymes and Domains as Ancient and Versatile Signaling Devices"

_ijms, 2024, doi:10.3390/ijms25179535_

Round 1

Reviewer 1 Report

Comments and Suggestions for Authors

It is a well described review regarding "Moonlighting crypto-enzymes" and a significant overview regarding this research field. 

Minor point:
There are plenty of self-citations (about 50% of all citations). Please check whether all of these self-citations are necessary.

Author Response

Reviewer: It is a well described review regarding "Moonlighting crypto-enzymes" and a significant overview regarding this research field. 

- We appreciate the assessment.

Minor point:
There are plenty of self-citations (about 50% of all citations). Please check whether all of these self-citations are necessary.

- First, we affirm that the citations are essential for the argument we present. Second, in response to the comment, we have also added > 30 new citations. 

Reviewer 2 Report

Comments and Suggestions for Authors

The manuscript would very much profit from more comprehensive graphic work. There is nothing wrong with the choice of graph types, but they could be more pleasing to the readers eyes. Also the section on how many hits on known AC and GC domain proteins  the different approaches yielded would be nice. Does for example non-identification (of a known AC or GC) mean that the catalytic domains are more variable than expected or that the search algorithm is not optimized enough?

Minor

page 11, line 401 strain is meant, I guess

I am looking forward to more such approaches, however, my impression lately was that it stagnated. The great paper of Mirabeau et al, 2007, Genome Research 17: 320-327 on a hidden Markow model search for human peptide hormones not only confirmed almost all known peptide hormone but also predicted successfully many more that were found afterwards.

Author Response

The manuscript would very much profit from more comprehensive graphic work. There is nothing wrong with the choice of graph types, but they could be more pleasing to the readers eyes. Also the section on how many hits on known AC and GC domain proteins  the different approaches yielded would be nice. Does for example non-identification (of a known AC or GC) mean that the catalytic domains are more variable than expected or that the search algorithm is not optimized enough?

- We would argue that the graphic works is both sufficient and of good quality to make the arguments we put forward. 
The reviewer's comment about the number of hits with modified motif sequences is insightful and indeed in several of the cited publications we have listed all hits with modified motifs in the "supplementary material" files. We therefore believe that including these numbers in this review would only distract from our central arguments.
The comment about the optimisation of the algorithm is valid too. We state and imply that our motif modifications are based on rational substitutions followed by experimental testing - we are not able to predict ALL active sites, but our concern is first and foremost to avoid false positives. Whether further computational optimisations can be achieved remains to be seen.

Minor

page 11, line 401 strain is meant, I guess

- corrected

I am looking forward to more such approaches, however, my impression lately was that it stagnated. The great paper of Mirabeau et al, 2007, Genome Research 17: 320-327 on a hidden Markow model search for human peptide hormones not only confirmed almost all known peptide hormone but also predicted successfully many more that were found afterwards.

- We concur and would like to add that our approach is not based on a hidden Markow model but on analyses of amino acids essential for biochemical function and structural conditions in reactive centres. 

Reviewer 3 Report

Comments and Suggestions for Authors

Your article is interesting but some modifications are recommended.

1. Cryptic moonlighting enzymes in plants and animals. I suggest to summarize these too much information in a table form.

2. In "Structural and functional aspects of cryptic mononucleotide cyclases", you referred to the cytosolic caclcium but the point and the relation is not clear. Would you add some details.

3. Why you focused on plant proteins rather than animals' proteins?

4. Fig. 1 is your own or referenced from elsewhere? and the same for the rest of the figures because Fig.2 is referenced.

5. In Fig. 1, what does RKS, YEW, x 8, 11.... etc mean? and the same for the rest of the figures. please clarify the meaning in the figure ligand.

6. Too much self-citations, which could be an ethical concern.

Author Response

1. Cryptic moonlighting enzymes in plants and animals. I suggest to summarize these too much information in a table form.

- We do not think that key message - moonlighting proteins and how to find them - does lend itself easily to tabulation.  

2. In "Structural and functional aspects of cryptic mononucleotide cyclases", you referred to the cytosolic caclcium but the point and the relation is not clear. Would you add some details.

- The role of calcium in nucleotide cyclase activity is extensively covered in several articles that we cite, e.g. :
Muleya, V., Wheeler, J.I., Ruzvidzo, O., Freihat, L., Manallack, D.T., Gehring, C. and Irving, H.R.
Calcium is the switch in the moonlighting dual-function ligand-activated receptor kinase phytosulfokine receptor 1.
Cell Commun. Signal. (2014) 12, 60. 

3. Why you focused on plant proteins rather than animals' proteins?

- One reason for the choice of plant systems is historical, since we pioneered our search approaches to discover the long elusive nucleotide cyclases in plants. In addition, plant transgenics are easily obtained and this has experimental advantages. However, we have also discovered novel animal cyclases e.g. 
Turek, I., Freihat, L., Vyas, J., Wheeler, J., Muleya, V., Manallack, D.T. Gehring, C. and Irving, H. 
The discovery of hidden guanylate cyclases (GCs) in the Homo sapiens proteome.   
Comput. Struct. Biotech. J. (2023), 21, 5523-5529; doi: 10.1016/j.csbj.2023.11.005

Finally, all computational and in vitro methods we apply are in use in plants and animal systems.

4. Fig. 1 is your own or referenced from elsewhere? and the same for the rest of the figures because Fig.2 is referenced.

- All figures, including the referenced ones have been drawn by the authors themselves.

5. In Fig. 1, what does RKS, YEW, x 8, 11.... etc mean? and the same for the rest of the figures. please clarify the meaning in the figure ligand.

- The letters in square brackets [] represent different amino acids allowed in a position of the motif, the round brackets () indicate the gap sizes. This statement was added to the Legend.

6. Too much self-citations, which could be an ethical concern.Reviewer 

- We have since included > 30 citations from "other" authors.